

# A GIS-based monitoring and early warning system for cover-collapse sinkholes in karst terrane in Wuhan, China

Li Xueping[1], Xiao Shangde[2], Tang Huiming[1], Peng Jinsheng[2]

[1]Faculty of Engineering,China University of Geosciences, Wuhan, 430074, PRC

[2]Hubei Province Geological Environment Terminus, Wuhan, 430051, PRC

*Correspondence to*: Li Xueping (lixp@cug.edu.cn)

**Abstract:** To reduce disastrous losses caused by karst collapse especially in urban areas, it is important to establish an early warning system utilizing monitoring data. Three major aspects have been monitored based upon engineering geological conditions and characteristics of karst collapse processes in Wuhan, China: changes in surface soil, soil deformation, and groundwater levels. Measurements have been recorded of: (1) soil pressure, (2) ground-penetrating radar images, (3) underground water levels, (4) ground water levels, (5) rainfall, (6) cracking, (7) ground deformation, and (8) water level in monitored wells. This paper has selected geological radar cross-sectional data and underground water level monitoring data to obtain criteria for hydraulic gradient warning, geological radar warning and plastic zone warning based upon these monitoring data and wider knowledge of karst collapse in Wuhan. A comprehensive warning system has been developed on a MAPGIS platform, employing monitoring data in Microsoft Excel format and Microsoft Visual C++ development tools. Three warning levels are adopted by the system: "safe", "becoming dangerous", and "dangerous"; indicated in green, yellow and red respectively on hazard maps. The system automatically undertakes processes of data management and model calculation leading to geo-hazard warning map generation. Using monitoring data collected in the first six months of 2011 at Wuhan, the system has established a hydraulic gradient model, plastic zone warning model, geological radar warning model, and a comprehensive early warning model; and has been shown to be an effective method of providing karst collapse warning.

**Keywords:** collapses and sinkholes; monitoring; early warning; GIS; Wuhan

## 1 Introduction

Cover-collapse sinkholes refer to sudden ground deformations in loose soil above solution cavities caused by external forces or human activities, resulting in the formation of sinkholes in many situations. Cover-collapse sinkholes are common all over the world and have been documented in many karst regions of the world (De





Bruyn and Bell, 2001; Farrant and Cooper, 2005; Gutiérrez et al., 2007, 2014; Currens, 2012; Parise et al., 2015 ) and usually cause engineering problems and significant economic losses, even involving personal damage (Martin, 1995; Boyer, 1997; Waltham, 2008). Since cover-collapse sinkholes tend to occur suddenly and their locations tend to be hidden, their study has become an important research topic aiming to develop an effective

prediction and early warning system.

The majority of cover-collapse sinkholes are distributed in the southwest, south, central and north parts of China, although some are scattered in other regions as well. Wuhan, the capital city of Hubei Province, is located in central China between longitude 113 ° 41'-115 ° 05 ' E and latitude 29 ° 58'-31 ° 22' N (Figure 1). Recently, with rapid economic development and urban area expansion, more buildings have been constructed on karst terrane

and surrounding areas; furthermore, other human activities, for example, underground water extraction and overloading have rapidly changed the geological environment, often causing cover-collapse sinkholes. This has seriously impeded the development of the city and threatened the safety of its residents. Figure 2 shows cover-collapse sinkholes in Fenghuo village, Qingling Xiang, and Hongshan District of Wuhan City present that appeared on April 6, 2000. The sinkhole in Figure 2 covers about 1,000 square meters with a maximum depth of

10 metres, and nearly one thousand villagers were adversely affected. The collapses shown in Figure 3 (a) and (b) occurred in the Fasi town, Jiangxia districts of Wuhan on September 5, 2014. Nine cone-shaped sinkholes of various sizes were found on the site.

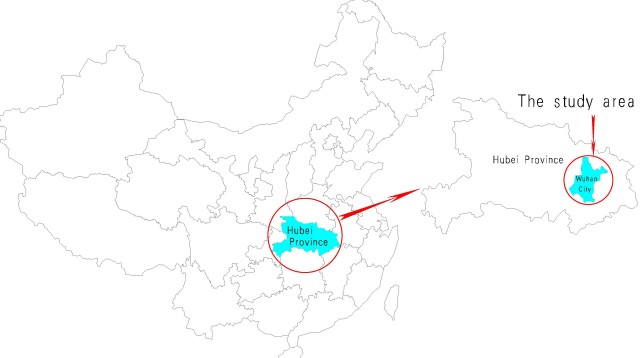

Fig.1. Location of Wuhan in China





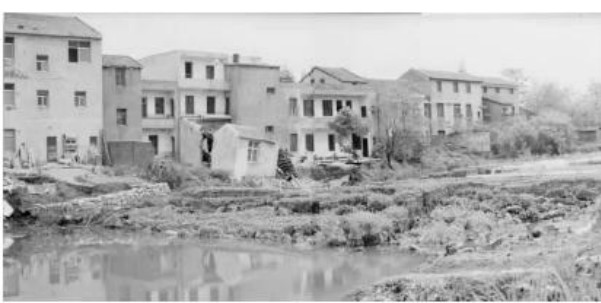

Fig.2. Damage scene caused by the cover-collapse sinkholes in Fenghuo Village in 2000

Source: Shen (2014)

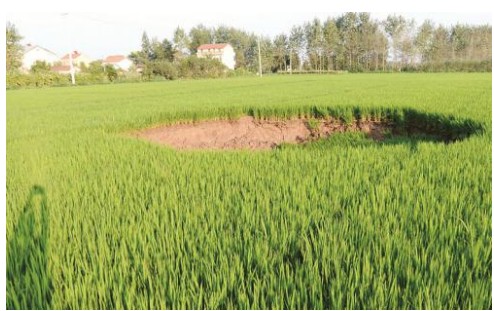 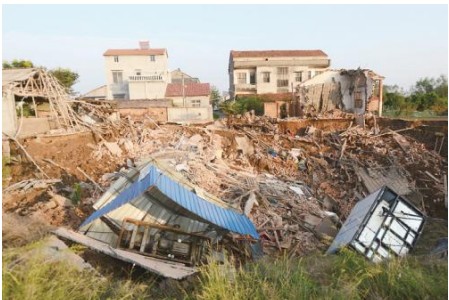

Fig.3. (a) Cover-collapse sinkhole in Fasi town;          (b) Karst collapse in Fasi town in 2014

Source: Jinchu Net (2014)

In order to prevent and mitigate the geo-hazard caused by cover-collapse sinkholes, the local government has carried out some work on the monitoring in an attempt to provide early warning.

Geographic Information System (GIS) technology has been usually uesd to build karst databases (Cooper et al.,

2001; Lei et al., 2001; Green et al., 2002; Gao et al., 2005a, 2005b; Wu et al., 2008) and assess for cover-collapse sinkholes vulnerability & risk (Yilmaz, 2007; Cooper, 2008; Koutepov et al., 2008; Li et al., 2011; Lamelas et al., 2011; Ozdemir, 2015). The main objective of this paper is to describe an early warning system employing a GIS technique developed to fulfil this prediction purpose. The system has been implemented and tested in Wuhan.

**2 Methods of monitoring cover-collapse sinkholes**

The two most commonly used monitoring methods of cover-collapse sinkholes can be categorised as deformation monitoring and underground water monitoring (Li et al., 2005). The former is achieved in two ways:

(1) directly monitoring deformation of the ground surface or underground (e.g. observing land surface



subsidence, land surface cracking and building cracking etc.); (2) monitoring underground soil deformation using ground-penetrating radar (GPR), fibre optics, and other methods (Meng et al., 2011; Intrieri et al., 2015). The latter includes monitoring the dynamic underground water level and water or gas main pressure changes in karst terrane (Lei et al., 2004).

## 2.1 Ground-Penetrating Radar (GPR)

Ground-Penetrating Radar is an indirect detection method using electromagnetic (EM) radio waves (1 MHz ~ 1 GHz) to discover the distribution of underground structures. The basic principle of GPR is that EM waves are radiated as high-frequency signals into the ground through a transmitter (T); another receiver (R) is used to receive diffracted and refracted signals, direct signals and interfered waves. The received signals can be processed to indicate the distribution of underground materials.

GPR is widely used to locate karst hazards such as cavities and paleocollapses and to identify their characteristics (Collins, et al., 1990; McMechan et al., 1998; Chamberlain et al., 2000; Nouioua, 2003; Prokhorenko, et al., 2006; Mochales et al., 2007; Pueyo-Anchuela et al., 2009; Leucci et al., 2010; Gómez-Ortiz et al., 2012; Vadillo et al., 2012). Disturbed cave soils have distinctive dielectric properties when compared with pristine soils in the surrounding area, so by contrasting cross-sectional maps of the same traverse collected regularly over time by GPR, it is possible to estimate underground soil movement, and this helps to monitor the cave in terms of its formation and development, and thus enables prediction of cover-collapse sinkholes.

## 2.2 Dynamic underground water level monitoring

The impact of a cover-collapse sinkhole on underground water is mainly reflected by water table dropping and fluctuation. Frequent changes in the underground water table can damage or rearrange the ground. ('Water table' is a technical term for underground water level.) When the water or gas pressure fluctuates or the water pressure at the bottom of the cover layer reaches a critical value, the cover layer will be damaged and then karst collapse occurs. The main task of dynamic underground water monitoring is to identify the dynamism of underground water level or levels in the studied area and analyze the hydraulic connection between aquifers. By monitoring changes of groundwater level, cover-collapse sinkholes can be predicted (Feng et al., 2007).

## 3 Characteristics of cover-collapse sinkholes and monitoring methods in Wuhan

### 3.1 Characteristics of cover-collapse sinkholes





### 3.1.1 Geological background

Wuhan is located in the middle and lower reaches of the Yangtze River, and the landscape includes the Yangtze River, its flood plain and that of a major tributary the Hanjiang River, accompanied by terraces, and ridge-hillock terraces above 100m sea level. The bedrock in Wuhan is Silurian to Quaternary in age.

Carboniferous limestone or dolostone layers are as much as 24 - 34 m thick; Permian limestone is 170m; while Triassic limestone and marl is about 410 m especially in the core portion of a syncline. Quaternary sediments are generally 30 - 40 m thick with a maximum of 65 m, mainly distributed along the Yangtze and Hanjiang Rivers. The lithology is clayey, sandy loam, sand and gravel, etc.

### 3.1.2 Distribution of limestone

There are three main limestone bands distributed east-west across the Yangtze River in Wuhan from north to south, and the remaining area of Wuhan is scattered with linestone (Fan, 2006). As can be seen in Figure 4, the first band (brown) is located in the northern part of the city (on the left bank of the Yangtze River), and from Dai Jiashan to Zhanjiaji, and Jiangjiadun to Wuhan Iron and Steel Group Corporation works (on the right bank of the Yangtze River). The length of this band is about 17 km from east to west and its width is 0.8 km - 2.8 km.

There have been no recorded karst collapses within this band.

The second band (pink) is located in the middle of the city. It starts from Zoujiawan in Hanyang, through Shilipu to Guishan (on the left bank of the Yangtze River), then Sheshan - Wuhan University - Mayizhang (the right bank of the Yangtze River). The length of the second band is about 35 km from east to west and its width is 0.5 km - 2.0 km. There are no recorded cover-collapse sinkholes within this band either.

The third band (purple) is located in the south of the city, from Taizhi Lake in Hanyang in the north and the left bank of the Yangtze River, through Lujiajie and Wutaiza in Wuchang to the Nanhu area. The length of this band is about 35 km from east to west and its width is 4.0 km - 5.0 km. This is the widest band of limestone in Wuhan, and all of the recorded collapses have occurred within this band.




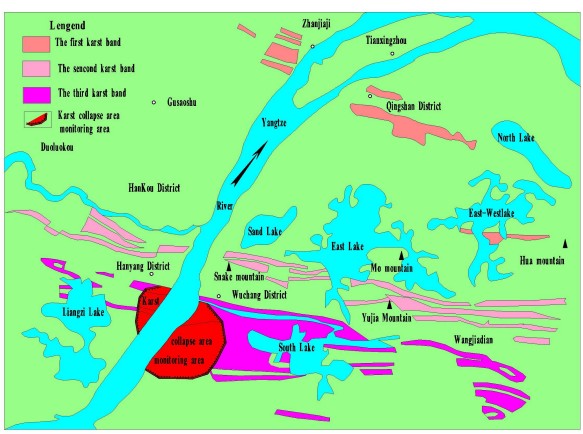

Fig. 4. Limestone distribution bands in Wuhan.

Source: Fan (2006)

### 3.1.3 Characteristics of karst collapse in Wuhan

In the Wuhan area, the vertical depth of karst terrane ranges between 27.86 and 176.19 m, but the majority is

less than 100 m. Karst caves are normally present at a depth of 30 m; in terms of size, their maximum diameter

is 11 m, minimum is less than 0.l m; most of these caves are distributed along east-west trending syncline axial

troughs and anticline crests (Fig.4).

The earliest recorded karst collapse in Wuhan, the Ding Gongmiao karst collapse, occurred in August 1931.

Since 1978, collapses of various sizes have been reported in both Hanyang and Wuchang districts in Wuhan;

including collapses at the Hanyang Steel Factory, Ruanjiaxiang in Baishachau, Lujiajie Secondary School,

Maotangang Elementary School, Tujiagou Law School and Fenghuo Village.

The karst collapse area of Wuhan is located in the middle and front of the first terrace of the Yangtze River

where the overlying soil is a Quaternary Holocene loose accumulation layer ($Q_4$) with a thickness of 5-25 m.

Soil caves exist in the soil layer, and the soluble carbonates karst has developed in the underlying soil. Open

karst caves (Semi-filling or no filling) form at the top of the soil layer which become the channels and spaces for

transporting and storing a large amount of potential corrosion. Due to the changes of the hydrodynamic

conditions and the additional loading, soil collapses appeared in the $Q_4$ or karst caves as consequences of cave

roof collapse.

Based upon determinants of the karst collapse in Wuhan, Zhao et al. (2012) selected the karst foundation

conditions (karst stratum, karst distribution type, karst development levels, the influential degree of geological





structure), upper cover conditions (thickness and structure of the covering layer), and hydrogeological

conditions (the relationship between pore water and karst groundwater, groundwater fluctuation, distance from

the Yangtze River, whether it is affected by groundwater exploitation), which include three factors with a total of

ten indicators as the fuzzy hierarchy monitoring criteria in the karst collapse hazard model. This was used to

evaluate the risk of karst collapse in Wuhan, and the results are shown in Figure 5.

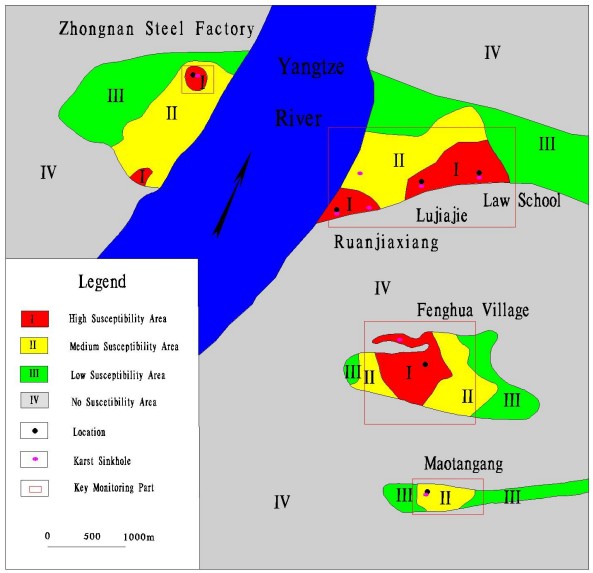

Fig.5.The risk distribution of karst collapse in Wuhan

### 3.1.4 Previous karst collapse research in Wuhan

Based upon case studies of karst collapses in Wuhan Zhongnan Steel Factory, Liujia Street, Maanshan Oil Field,

Yingwu Island in Hanyang, and Doukou Lake in Wuchang, empirical studies examined the mechanism of

collapse based on the geological conditions in the above areas, and proposed countermeasures during

construction (Liu, 1982; Li, 1989; Xiao, 1990). Studies have also offered in-depth discussion with regard to the

mechanism causing cover-collapse sinkholes in Wuhan (Liu, 2001; Zheng, 2003; Fan, 2006; Hu, 2007).

Guan et al. (2008) summarized the spatial characteristics of karst distribution in Wuhan, stating that karst is

relatively shallow and the sizes of caves are small. GIS technology and a partial least squares path model (PLS)

were applied to assess the karst collapse risk in Wuhan (Feng, 2008). A karst collapse risk assessment model

using an analytic hierarchy process (AHP) method was constructed to carry out risk classification and evaluation



in Rujiaxiang and Lujiajie (Zhao et al., 2012). A Kriging spatial interpolation method based on drilling data was used to study the cover karst development pattern with the assistance of GIS software (Tu et al., 2014). The karst collapse risk level in Wuchang, one of the three districts of Wuhan City, was assessed by using fuzzy algorithms and modular GIS technology (Zhong et al., 2015).

Using Wuchang as a case study, Lei et al. (1993) showed the mechanism by which collapse developed and major antecedents of cover-collapse sinkholes using a large-scale physical model. Jia et al. (1994) employed a combination of GPR data and a numerical simulation method, carried out a stability calculation and attempted to predict potential areas of karst collapse in Wuhan.

Extant literature has provided insightful understanding on karst collapse in Wuhan, and has helped to reduce
losses caused by karst collapse. With the rapid economic development in Wuhan in recent years, it has become the most populous city in Central China, with population density of 1,152 per square kilometre; and the sixth populated city in China in 2015 (Nations Online Project, 2016). Meanwhile, the number of karst collapse has also increased in numbers and has lead to greater losses, especially, in built-up areas. The urgency to manage this hazard has been brought to urban planning agenda. This article intends to utilise multiple-source of
monitoring data collected in various formats in Wuhan; to develop an early warning system suits Wuhan; and attain timely warning generated by an automatically process. This research can certainly benefit karst collapses management in other areas in China.

### 3.2 Monitoring karst collapse in Wuhan

A project, "Wuhan cover-collapse sinkholes geo-hazard monitoring and early warning" was undertaken by the
Hubei Geological Environment Station (HGES) as part of the project "Typical geological disaster monitoring and early warning and demonstration project", sponsored by the China Institute of Geo-Environment Monitoring (CIGEM) in 2006. By the end of 2008, HGES had established a cover-collapse sinkhole monitoring network in the high susceptibility karst collapse area, and has been monitored collapse since then. The selected monitoring area is shown in red in Figure 4. Key monitoring areas are shown in Figure 5. According to the characteristics of
karst collapse in Wuhan City, the karst collapse monitoring and early warning technology is mainly built on three aspects: rock and soil movement, building deformation and groundwater change, together with analysis of geologically environmental conditions. The monitored content (Shen et al., 2014) was as follows:

(1) Soil change monitoring

Two methods were used to monitor soil change: examining pressure meters buried in the ground in order to




monitor pressure; tracking vibrating layers and cave cross-profiles from scan cross sectional profiles obtained by

the GPR.

(2) Deformation monitoring

Deformation monitoring included land subsidence monitoring and land surface crack monitoring.

(3)  Groundwater monitoring

This included tracking groundwater levels in monitoring holes and wells. The former further includes examining

the groundwater tables in Quaternary pore water and karst water.

**3.3 Cover-collapse sinkhole monitoring and warning implementation**

In order to monitor and to provide early warning of cover-collapse sinkholes, there were four critical elements:
management of all types of monitoring data, development of warning criteria, measurement of severity of
warning and warning classification.
(1) Warning classification
Three levels were used to classify the warning grade. Level I is "safe" indicated as green on the warning map;
level II refers to the state "becoming dangerous", yellow; and level III "dangerous", red.
(2) Warning cell size
The cover-collapse sinkhole monitoring area was gridded at 50m × 50 m as warning units in Wuhan, with the
warning signal shown in each unit in green, yellow and red colour accordingly. Grid units outside of the
monitored area are displayed in light blue.
(3) Monitoring and warning system implementation
(4) The monitoring and early warning cover-collapse sinkhole system was further developed on a GIS software
platform.

**4 Selection of warning criteria**

Based upon the cover-collapse sinkhole monitoring project and early monitoring knowledge, the current
warning system selects data collected through GPR and underground water levels to determine the warning
classification.

**4.1 Hydraulic gradient warning criteria**

The decline of karst water level in the study area leads to the soil destruction and the karst collapse. According

to the monitoring data, the Yangtze River water (surface water) and the groundwater in the study area show

clear a hydraulic connection. In the rainy season, the Yangtze River water supply the underground water, and

the underground water level rises with the water head reaches up to 7m. In the dry season, groundwater is

discharged to the Yangtze River in which the groundwater fluctuates about 3m, and the water level of the

Yangtze River fluctuates up to 15m. Therefore, the hydraulic gradient can be calculated according to the water



level and the height between sand roof and carbonate top. By comparing the calculated hydraulic gradient and the critical hydraulic gradient of soil failure, the risk assessment of karst collapse can be achieved.

Using water level data of Quaternary pore water and karst water collected over the same period of time, it is possible to calculate the hydraulic gradient of each warning grid and thus the likelihood of karst collapse can be

predicted according to the impact of hydraulic gradient on karst collapse.

Lei et al. (1993) used a physical model of the first terrace of Yangtze River sediment in Wuchang and determined the mechanism of soil collapse derived from karst leakage deformation caused by falling water level corroding limestone bedrock. In their simulation, the critical collapse gradient for the surface layer as a single structured clayey soil was 12.89; and for a double surface layer with a dual structure of clayey soil (clayey upper

part and sandy soil lower part) was 2.03. Therefore, the hydraulic gradient warning criteria in this paper are designed as follows: for a cover layer as a single structured soil, the hydraulic gradient for warning level III is taken as 10.3 and level II is 7.7 (80% and 60% of the experimental data 12.89 respectively); for a dual structured covering layer of soil (clayey and sandy soil), level III is 1.6 and level II is 1.2 (80% and 60% of the experimental data 2.03 respectively).

**4.2 Warning criteria in a plastic zone**

**4.2.1 The warning method for plastic zones**

The processes to establish the warning criteria in the plastic zone are as follows: using FLAC3D software to establish 3D geo-mechanical models in areas where with the presence of karst sinkholes or abnormal soil confirmed by physical exploration and soil holes using a drilling survey. Below the water table in the phreatic

zone where the expansion of solution cavities as the plastic zone is gradually excavated causes cavern collapse, it is possible to analyze numerical simulation results to discover the location of a plastic area in the area surrounding a soil cave, and changes in location and strain in the soil surrounding the cave. Based upon these changes in the plastic zone, it is possible to forecast the occurrence of cover-collapse sinkholes.

The warning criteria for the plastic zone in the system are as follows: with a normal underground water table, if

the plastic zone extends to ground level, the grid square is shown with a level III warning; if the plastic zone did not extended to the ground level, but shows extension compared with the simulated result at previous time, i.e. the zone depth becomes shallow and its area increases, the grid is shown as a level II warning; all other grids are shown as level I warning.



### 4.2.2 A case study of plastic zone monitoring

In April 2000, a karst collapse in Fenghuo village generated 22 sinkholes; the largest one was 52 m in length, 33m in width, with a depth of 7.8 m in the deepest place. It caused collapses of 42 buildings; affected 230 households and led to the relocation of 990 people. It also destroyed large agricultural land, though no death was

reported. The direct economic loss is about 6 million (RMB) and indirect loss about5 million RMB, making it the worst karst collapse in the recorded karst collapse history in Wuhan ( Fan, 2006 ). The village has become one of the key monitoring areas; we then selected this area as the case study of the plastic zone monitoring. The derivation of warning criteria using the case of Fenghuo village as follows:

(1) 3D geological model

The bedrock in the case study area is Triassic carbonates. Drilling data shows the cover layer is Holocene fluvio-lacustrine deposits on the first terrace of the Yangtze River, which has a dual-layered structure with a thickness of 24-28 m. The upper part of the cover layer is mainly yellow-gray clay, loam, occasionally with silt; it is damp and plastic with a thickness at 3-6m; the lower part is gray - dark gray silt sand with a thickness at 22-24m.

(2) Figure 6 shows generalized 3D geological model.

The model dimensions are 10m long, 5 m wide, 6 m high.

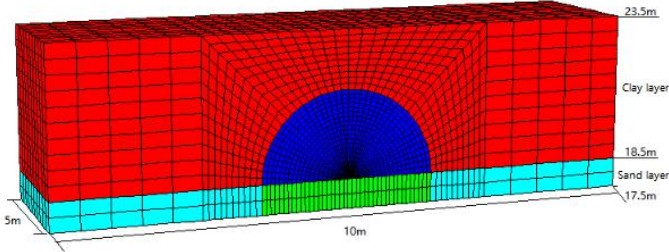

Fig. 6. Generalized 3D geological model for case study in Fenghuo Village

(3) simulated conditions

Table 1 shows underground water level monitoring data of the simulated area in 2011. The monitoring point of karst water level is CK16 inner, and of pore water is CK16 outer.

Table 1 the underground water level monitoring data

| Monitoring | Underground water level altitude (m) | | |
|---|---|---|---|
| Points | Previous | Last | Difference |
| CK16 inner | 17.28 | 17.14 | -0.14 |





| CK16 outer | 18.50 | 19.48 | 0.98 |
| --- | --- | --- | --- |

The pore water table level is selected in this case study because the karst water level is above the karst bedrock roof (i.e. the karst cavity is in the phreatic zone). The simulated conditions are:

a) During the summer wet season: groundwater level is 19.5m.

b) During normal seasons: groundwater level is 18.5m.

5  c) During the dry winter season: groundwater level is less than 17.5m.

(4) Simulation results

Simulation results at different underground water table levels are in Table 2; and the plastic zone distribution around the soil cave at different groundwater conditions are shown in pink in Figure 7.

Table 2 The simulation results at different underground water levels

| Groundwater Level /m | Plastic zone Depth /m | Max stress low value | Max subsidence /cm | Soil cave state |
| --- | --- | --- | --- | --- |
| < 17.5m | 1.9 | Exist | 4.7 | Stable |
| 18.5 | 1.3 | Exist | 7.8 | Stable |
| 19.5 | 0.6 | Exist | 11.3 | Stable |

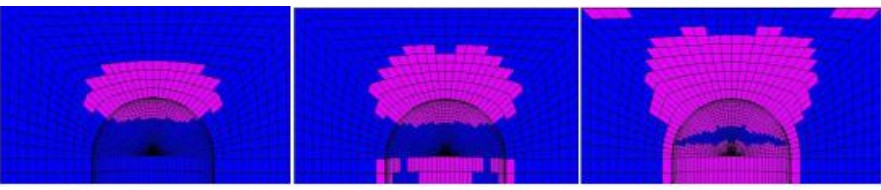

**7** (a) water level less than 17.5m      7 (b) water level at 18.5m      7 (c) water level at 19.5m

Fig.7. Plastic zone distribution (in pink) surround a soil cavity under different groundwater levels

(5) Early warning test results

15  The simulation results show that the extent of the plastic zone has expanded compared to previous estimates, but has not reached ground level under the test water level conditions; and therefore the warning level for grid square No.4088has been set at level II and displayed in yellow on the warning map.

**4.3 Warning criteria from GPR monitoring data**

The GPR data reveals that the abnormal depths, abnormal shapes and sizes of underground structures are

20  different compared with previous monitoring results, the warning criteria are:

(1) When there is a sign of expansion compared with previous monitoring data, to be specific, abnormal depths become shallower, or their size becomes larger, the abnormal area in a grid square is classified as a level III



warning.

(2) If there is no significant change, the abnormal area of the grid is classified as the level II.

(3) If the GPR data shows no sign of abnormality, the related grids are classified as level I warning.

## 5 System Design

### 5.1 System Analysis

#### 5.1.1 System Requirements
Because karst collapse occurs suddenly and its location is likely to be hidden, it is usually monitored by multiple methods to mitigate geo-hazard, which tend to generate large amounts of data in various formats. The challenge to deal with such large data efficiently and fast is urgent and essential to establish comprehensive simulation models to provide the early warnings.    GIS has the capacity to store massive amounts of information, to search fast, to process data timely, and to manage dynamic information. It thus guarantees sustainable development in a karst warning system.

#### 5.1.2 Data Source Analysis

(1) Monitoring data

Techniques used to monitor cover-collapse sinkholes in Wuhan include automatically collected monitoring data, e.g. rainfall monitoring, and also data collected manually. The data covers

a) Soil pressure monitoring data;

b) GPR monitoring data;

c) Underground water level monitoring data;

d) Ground water level monitoring data;

e) Rainfall monitoring data;

f) Crack monitoring data;

g) Ground deformation monitoring data;

h) Water levels in observation wells

(2) Graphic data

Graphic data may include geographical information, lithology, geological structure, hydrogeological conditions, ground collapse sites and drilling site information. Graphics data mainly consist of the following three elements:

a) Zero-dimensional points representing spatial locations. For example, map location of, geological points and drill hole sites.

b) Lines representing 1-dimensional features with length and direction, such as roads, coastlines, hatching and





contour lines.

c) Polygons representing two-dimensional areas, such as distribution of strata, surface water and a variety of other geological areas.

**5.2 The system Design**

**5.2.1 System objectives**

The main data sources of cover-collapse sinkhole monitoring and early warning are based upon karst geological engineering survey data and karst surface monitoring data, leading to the recognition of soil cavities collected through GPR detection and the impact of underground water level on the karst surface. These form the core sources of the software. Specific objectives of the software are as follows:

(1) To manage information of the monitoring points;

(2) To produce tables and graphic results of the monitoring points and search timely;

(3) To manage the warning criteria;

(4) To classify the warning areas.

**5.2.2 Cover-collapse sinkholes warning implementation process**

Step 1: to build up the cover-collapse sinkhole background database, including lithology, geological structure, hydrogeological conditions, the results of geophysical prospecting, drilling results, results of laboratory experiments and monitoring project information.

Step 2: to divide the monitoring area into 50 m × 50 m warning grid cells.

Step 3: to establish the monitoring database and manage the time-series of the monitoring data.

Step 4: to calculate the hydraulic gradient for each grid cell based upon the underground water level monitoring data at each period of time; and then calculate the hydraulic gradient warning level for each cell according to hydraulic gradient warning criteria; and construct the hydraulic gradient warning level diagram.

Step 5: to calculate the distribution of plastic zones in simulation models using the underground water level monitoring data at each time point, and then assign a warning classification to each grid cell; draw the plastic

zone warning level diagram.

Step 6: construct a GPR warning level diagram using each set of GPR monitoring data to calculate the warning level for each grid cell;.

Step 7: Establish a comprehensive warning diagram throughout the monitored area based on warning levels of



the hydraulic gradient, the plastic zone and the GPR in each grid cell. The comprehensive warning criteria are as follows: (1) the warning level for the grid cell is level III if there is any level III warning of the above three attributes; level II if there is any level II existing in the above three attributes; otherwise the grid cell is level I. The coding rules of comprehensive warnings for each grid are shown in Table 3.

Table 3 the coding rules of the comprehensive warning in each grid

| Warning level | The first character hydraulic gradient | The second character plastic zone | The third character GPR |
|---|---|---|---|
| III | 3 | 3 | 3 |
| II | 2 | 2 | 2 |
| I | 1 | 1 | 1 |
| None grid cell | - | 0 | 0 |

Note for example, if an attribute value of a comprehensive warning grid cell is coded as 102, it represents a comprehensive warning level of II and the grid is displayed in yellow on the map. An attribute value of '102' refers to a hydraulic gradient of level I; with the plastic zone being outside the monitoring area; and the GPR 
being level II.

**5.3 Selection of the software development platform**

MAPGIS is tool-oriented GIS software developed by the China University of Geosciences (Wuhan) that provides a complete secondary development parameter library. The secondary development interface API (Application Program Interface) is a set of functions used by the application. By means of this set of interface 
functions users can develop, in a programming environment such as BORLAND C ++, VISUAL C ++, VISUAL Basic, the application-oriented GIS software for specific areas. Wuhan cover-collapse sinkhole monitoring and early warning system uses Visual C ++ and MAPGIS API as development tools.

The system uses MAPGIS vector data organizing methods, and vector map features are grouped into three categories: points, lines and polygons, according to their basic geometric features. There are three file types: 
point files (* .WT), line files (* .WL) and polygon files (* .WP). The monitoring data is managed in EXCEL format and can be directly inputted through a GIS interface or imported from EXCEL into the system.

**5.4 Data structure design**

**5.4.1 Monitoring data**

The monitoring data in the system includes information from monitoring points and observations over time. The



data structures are shown in Tables 4 and 5.

Table 4 monitoring point data structure

| Field Name | Field Type | Field Length | Decimal places | Memo |
| --- | --- | --- | --- | --- |
| PointID | Integer | 6 | / | |
| Point Number | string | 20 | / | |
| Point Name | short | 2 | / | |
| Xcoor | double float | 15 | 6 | |
| Ycoor | double float | 15 | 6 | |
| Zcoor | double float | 15 | 6 | |
| Equipment ID | string | 30 | / | |
| State Code | short | 1 | / | |

Table 5 time-series data structure

| Field Name | Field Type | Field Length | Decimal Places | Memo |
| --- | --- | --- | --- | --- |
| Point ID | string | 20 | / | |
| Point Value | float | 6 | 2 | |
| Monitoring day | date | | | |
| Monitoring hour | integer | 2 | | |

### 5.4.2 Warning grid

The data structure of the system warning grid file is shown in Table 6.

Table 6 the warning file data structure

| Field Name | Field Type | Field Length | Decimal Places | Memo |
| --- | --- | --- | --- | --- |
| ID | integer | 6 | / | Code of grid cell |
| Xcoor | double float | 15 | 2 | Grid cell X coordinate |
| Ycoor | double float | 15 | 2 | Grid cell Y coordinate |
| Hydraulic gradient | double float | 15 | 2 | Grid cell hydraulic gradient value |
| Cover layer type | short | 2 | / | 1:single;2:binary |
| Hydraulic gradient level | short | 2 | / | 1:I safe;2:II towards danger;3:III danger |
| Plastic zone level | short | 2 | / | 1:I safe;2:II towards danger;3:III class; 0: not monitored |
| GPR abnormal planar level | short | 2 | / | 1:I safe;2:II towards danger;3:III danger; 0: not monitored |
| GPR abnormal depth level | Short | 2 | / | 1:I safe;2:II towards danger;3:III danger; 0: not monitored |





| GPR warning level | short | 2 | / | 1:I safe;2:II towards danger;3:III danger; 0: not monitored |
| Comprehensive level | short | 2 | / | 1:I safe;2:II towards danger;3:III danger; 0: not monitored |

Table 7 The area file of GPR interpretation data structure

| Field Name | Field Type | Field Length | Decimal Places | Memo |
|---|---|---|---|---|
| IDGPR | integer | 6 | / | Code of GPR detection point |
| ID | integer | 6 | / | Code of grid cell |
| XcoorGPR | double float | 15 | 2 | GPR detection point X coordinate |
| YcoorGPR | double float | 15 | 2 | GPR detection point Y coordinate |
| upper depth | double float | 15 | 2 | Distance between the ground level and upper part of the abnormal area |
| lower depth | double float | 15 | 2 | Distance between the ground level and lower part of the abnormal area |

**5.5 System warning modules**

**5.5.1 Hydraulic gradient warning module**

The upper cover layer structure type for each warning grid square is calculated based on drilling data, and the result is assigned into a "cover layer type" attribute field. The hydraulic gradient of each grid is calculated using the underground water level monitoring data and carbonate rock roof depth information revealed by drilling. The attribute value for 'hydraulic gradient level' in the grid file '.WP' is obtained according to the hydraulic gradient warning criteria.

**5.5.2 Plastic zone warning module**

Geological models can be established based upon historical sinkhole data and the distribution of karst terrane revealed by geophysical exploration and drilling. Numerical simulations can then be carried out using the underground water level monitoring data at each period, establishing the corresponding relations between the simulation model grid position and the warning level. A WP file, and warning level obtained from simulation

results is assigned into the corresponding grid attribute field of "plastic zone distribution warning level". Typical cross-sectional collapses are selected in the investigation; multiple fixed models are established and used to simulate the collapse.

**5.5.3 GPR warning module**





The GRP warning module is achieved by the following technical process:

(1) Initial values for the warning file

An initial value '1' is assigned into a warning '.WP' file in attribute fields of 'GPR abnormal planar level', 'GPR abnormal depth level' and 'GPR warning level' if the grids are located in the GPR monitoring area;

otherwise, the value '0' is assigned. This is further explained in Table 6.

(2) GPR abnormal warning level calculation

The previous value in the field 'upper depth' is deducted from the current GPR abnormal planar monitoring data (.WP); if the result is less than 0, the attribute value in the field "GPR abnormal depth level" is set to 3; if the value equals zero, the grid attribute value is 2, as shown in Table 7.

(3) GPR warning level calculation

The system can accommodate values in the warning grid.WP file of both "GPR area warning level" and "GRP depth warning", and compare values among them, then take the bigger value and assign it to the field "GRP warning level" (see Table 6 for more details).

### 5.5.4 The Comprehensive warning calculation model

The system can allow for the values in the warning grid.WP file of "GPR warning level", "plastic zone level" and "hydraulic gradient level". It takes the maximum value of these three fields, and assign it to the field "Comprehensive level".

## 6 System applications

We have thus established a Wuhan cover-collapse sinkholes monitoring and warning system for the
demonstration area by monitoring data provided by Hubei Province Geological Environment Monitoring Station.

### 6.1 Monitoring data management

Information from monitoring points is managed in EXCEL (.xlsx files) that include: soil pressure, land surface movement, cracks, karst water level, pore water level, monitoring points in wells, ground water level and
rainfall level. There are 8 tables in each file corresponding to the above monitoring aspects. The file also includes information on date and length of monitoring; this enables us to build up time series data as shown in Figure 8. In terms of data input, the system can import the whole monitoring database or a single morning



record. After data input, the system supports fast searches and can generate time trend graphs. Figure 9 shows a

menu of the monitoring data management system. Figure 10 shows the search function, and Figure 11 shows the

time-trend graph for monitoring point CK13 in the system.

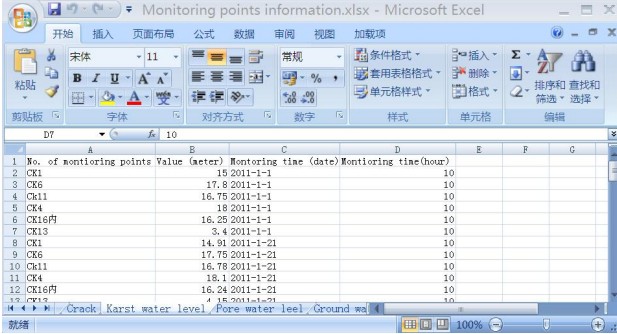

5                                   Fig.8. Information monitoring the database

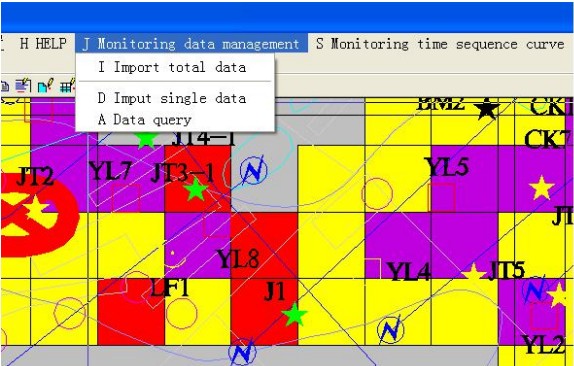

Fig.9. Menu monitoring data management




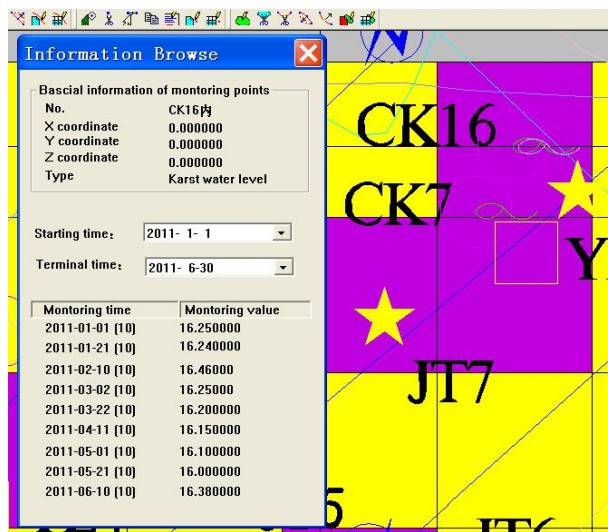

Fig. 10.    System data searches

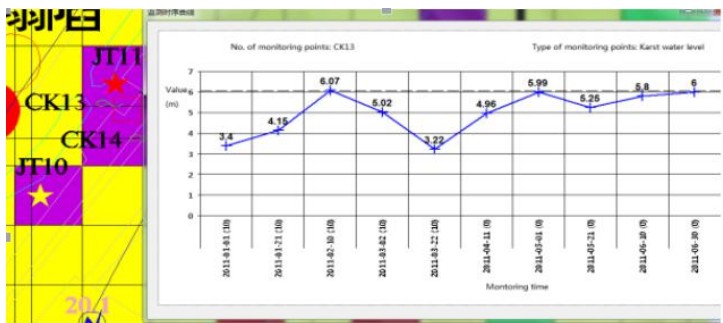

Fig.11. Time trend of monitoring point CK13

5    **6.2 Monitoring data warning level calculations**

**6.2.1 Hydraulic gradient warning level calculation**

The process for calculating the hydraulic gradient warning level is as follows:    first we calculate karst water

level and the pore water level in each grid cell based upon underground water level monitoring data; and then

we calculate the hydraulic gradient of each cell. We calculate the warning level of each cell in accordance with

10   hydraulic gradient warning criteria, and colour it accordingly. Figure 12 displays the hydraulic gradient warning

levels by using underground water level monitoring data from the first six months in 2011 in the case study area.





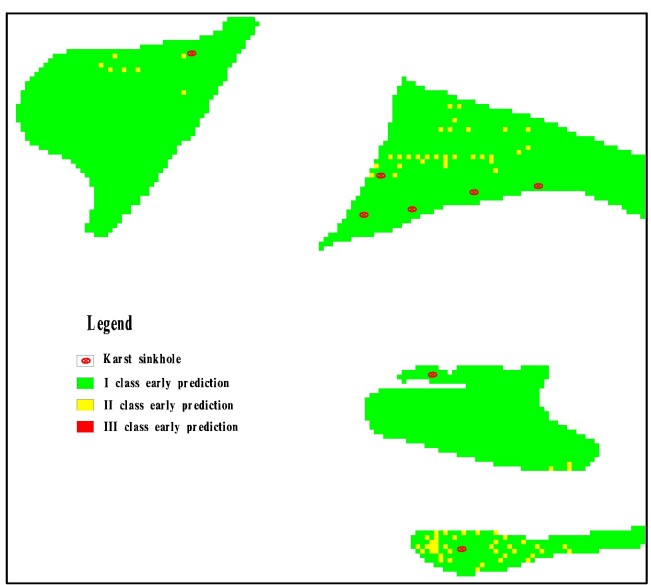

Fig.12. Coloured hydraulic gradient warning map

**6.2.2 Plastic zone warning level calculation**

Areas with detailed drilling data were selected using hydraulic gradient warning criteria and a number of
5    geological models were established.   Using underground water level monitoring data at each monitoring time
point, the distribution of plastic area areas were determined in each model; this provided information for
cover-collapse sinkholes warnings. The plastic zone warning level in Fenghuo Village was calculated based on
underground water monitoring data during the first six months in 2011, and is shown in Figure 13.

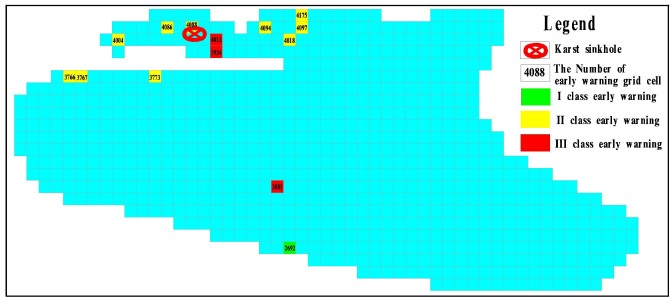

10                     Fig.13. 2011 warning levels in Fenghuo Village

**6.2.3 GPR warning level calculation**

The system is able to calculate warning levels based upon GPR monitoring data in 2011. Figure 14 shows the





GPR warning level in the Maotangang area.

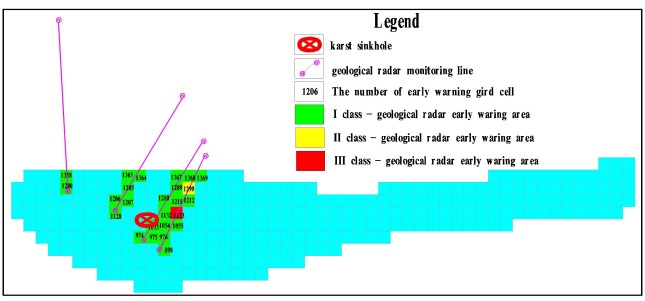

Fig.14. 2011 GPR warning level in Maotangang

### 6.2.4 Determination of comprehensive warning levels

Plastic zone warning levels and GPR warning levels were combined in a comprehensive warning level based on calculation results of hydraulic gradient warning levels. Figure 15 shows the comprehensive warning level in Maotangang.

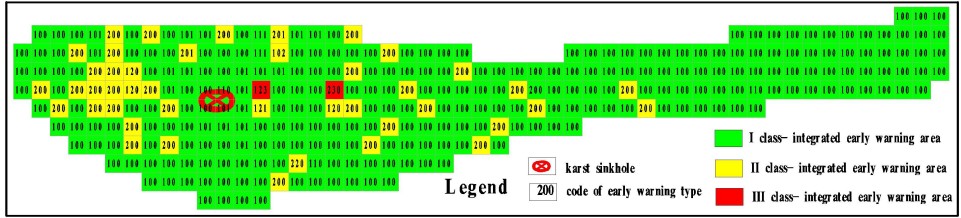

Fig.15. 2011 comprehensive warning level in Maotangang

## 6.3 Verification and countermeasures according to warning results

### 6.3.1 Hydraulic gradient warning

Figure 12 shows hydraulic gradient warning levels based on underground water level monitoring data in the first six months in 2011 in Wuhan. No warning level III appears on the map as can be seen from Figure 12, and there is no indication that penetrated deformation is unlikely to occur under normal underground hydrodynamic

conditions.

### 6.3.2 Plastic zone warning

Figure 13 shows a plastic zone warning level based on underground water level monitoring data in Fenghuo Village in the first six months in 2011. The warning level in grid cells 4012, 3934 and 3081 is level III.



Compared with historical data, the total fall extent of ground collapse in these three locations was less than 6mm. It is suggested that land fall monitoring intervals and ground level inspections should be increased to improve monitoring.

### 6.3.3 GPR warning

Figure 14 shows a GPR warning map of Maotangan based on GPR monitoring data in 2011. The GPR warning level is level II in grid cell 1250 and level III in grid cell 1130. Further verification was carried out through static electricity tests in the suspicious area, and graph generated verification of cone penetration tests showed no soil holes, but a relatively soft soil patch.

## 7 Conclusions and Suggestions

### 7.1 Conclusions

(1) The paper demonstrates the effectiveness of an extensive new monitoring system in providing early warning of sink holes in Wuhan, based on karst surface monitoring data and using established models (hydraulic gradient warning model, plastic zone warning model, geological radar warning model and the comprehensive early warning model).

(2) The application, developed with the technical support of MAPGIS, demonstrated its ability to manage cover-collapse sinkhole data, to analyze several cases and to generate warnings on maps.

(3) The system fulfilled its purpose in the demonstration area of Wuhan using monitoring data in the first six months in 2011 provided by the HGES.

### 7.2 Further suggestions for managing karst cover-collapse sinkholes in Wuhan

The accuracy of any early warning system is related to sinkhole surveying technologies, monitoring techniques, warning models and warning criteria. Karst surface monitoring in urban areas features a wide range of methods of investigation, vast investment, division of work, long monitoring periods, and fixed monitoring equipment that might be subject to vandalism. Here are our suggestions for improving cover-collapse sinkhole monitoring and warning in Wuhan:

(1) Continuous monitoring and database updating

Continuous monitoring and multi-source data can help to identify patterns of change in karst collapses; together with constantly updated databases they should be foundations to improve the accuracy of the warning system.





(2) Researching and establishing warning criteria in different types of area

The difference between warning system for a whole city and for one area is that the former is much bigger than the latter, and covers complicated karst engineering geological conditions. Furthermore, the main factors that cause the karst collapse differ between areas; this determines that various warning criteria may apply in the

same city. It is essential to establish independent criteria for different areas in order to improve the accuracy of karst collapse warning.

(3) Monitoring the impact of construction on cover-collapse sinkholes

In recent years, construction of viaducts, bridges and high-rise buildings with deep foundation has increased rapidly, and has caused cover-collapse sinkholes in Wuhan from time to time. The accidents apparently show an

increase in numbers. It is crucial to take into account potential consequences of construction activities when undertaking karst surface monitoring and early warning.

**Acknowledgements**

The research is funded by Wuhan Science and Technology Bureau key scientific and technological project (No. 201160823267).

The system has been applied for patent, and granted "computer software copyright registration certificate of the State Copyright Administration of the people's Republic of China" (No. 08044416).

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
