# Peer review of "A GIS-based monitoring and early warning system for cover-collapse sinkholes in karst terrane in Wuhan, China"

_Natural Hazards and Earth System Sciences, 2017_

## Referee Comment (RC1) · Anonymous Referee #1 · 6 Mar 2017

Dear editor and authors, I have had the opportunity to evaluate manuscript nhess-2017-22 submitted to "Natural Hazards and Earth System Sciences (NHESS)" entitled "A GIS-based monitoring and early warning system for cover-collapse sinkholes in karst terrane in Wuhan, China" and sunmitted by Li Xueping, Xiao Shangde, Tang Huiming, and Peng Jinsheng. Manuscript deals about an interesting subject; however there is a lack of coherence, order and systematic data description that do not permit its evaluation by a reader. Data is missing, the presentation is not ordered and context data are included along the whole manuscript from previous articles without indicating what is new in this manuscript. The conclusion of the model, without the consideration of the data, does not permit to know if interpretations are supported by data, producing

that manuscript is difficult to understand and does not have the scientific background, in this moment, for their evaluation to be published in a scientific journal. While further work is still needed, I suggest for the beginning some comments along the next lines related to the manuscript. Figure 1. A geological sketch is needed in order to locate the study area, a continental view can help for the location against only China surface. At figure 2 and 3, I assume that authors should have permission from both journals to include photographs from the cited manuscript, I suggest to editorial team to confirm this subject about the figure rights. The cities referenced where sinkholes has been identified should be included in the map from Fig. 1. The text from page 4 (paragraph from line 15) "so by contrasting cross-sectional maps of the same traverse collected regularly over time by GPR, it is possible to estimate underground soil movement, and this helps to monitor the cave in terms of its formation and development, and thus enables prediction of cover-collapse sinkholes." Requires some references where this subject has been previously applied. On the other hand, the evaluation of GPR in order to analyze karstic underground features should require to include, at least, some example of the obtained results and the compared evaluation of the same profiles during time to identify the changes related to the GPR record (besides the underground characteristics, soil state can also produce changes in the radargrams). About the dynamic underground water level monitoring is difficult to understand as a generalization that collapse are produced by the pressure increase or the term "the cover layer will be damaged", this subject requires explanation. About chapter 3.1.1. a geological map, section, borehole distribution or a cross-section should help in the interpretation of the context where the later analysis is carried out. If cross-section is included, the location of the water level should be also interesting to be included. Figure 4 also could require permissions from the journal from where it comes from, the aspect of the figure is not clear and besides the bands of carbonatic rocks, other geological subjects are required to be included. Moreover the bands are assumed to be limestones bands and not karstic bands as are referenced in the figure. Authors indicate that no evidences of karstic processes are in some of them. Moreover the reference to the structural

setting, fold axis for example, requires being included in the figure. The quality of fig. 5 is not evident, as there is not information that permits to contextualize it, moreover this does not make reference to risk, if there were anything evaluated in this figure should be hazard, peligrosity or susceptibility (not risk distribution). At figure group IV is incorrectly written and there is no sense about what "no suscetibility" means. The rest of the manuscript include description of what is presented to be done, with the considerations of the conclusions but there are not a data integration, comparison or discussion different than the general description. I think that what is presented can be of interest for the community but further work is needed including presentation, description and discussion of data. In this moment is more a general report of what to do, what can be expected and some results without the possibility to be evaluated by readers. In this sense, I suggest to modify significantly the manuscript before further revision.

---

## Author Comment (AC1) · 9 Apr 2017

Responses to reviewer 1's comments
Ref: nhess-2017-22

We appreciate the time taken to review this paper and constructive comments were given, we have made amendment accordingly. We hope that the editor and reviewer 1 agreed with our assessment that the paper has been substantively improved.

*1. however there is a lack of coherence, order and systematic data description that do not permit its evaluation by a reader. Data is missing, the presentation is not ordered and context data are included along the whole manuscript from previous articles without indicating what is new in this manuscript. The conclusion of the model, without the consideration of the data, does not permit to know if interpretations are supported by data, producing that manuscript is difficult to understand and does not have the scientific background, in this moment, for their evaluation to be published in a scientific journal.*

This paper intends to introduce the research and development of a GIS-based monitoring and warning system for cover-collapse sinkholes in karst terrane in Wuhan. To achieve an early warning, first, it is to carry out monitoring works. The selection of monitoring sections and monitoring proposals is based on the study of karst geologic conditions in the study area; the choice of monitoring method is mainly based on its credibility. And then we can apply for the early warning process which is involved in the research of the early warning criteria. Therefore, the main structure of the paper is shown as figure 1:

[Figure]

Figure 1 The main structure of the paper

The main focuses of the paper are in session 4: Selection; Session 5: System design and Session 6: System application. The analysis of karst geologic conditions in Wuhan is based on peer reviewed research. The main contributions of this research are shown in the Figures 5-15.

Based on the study of karst geologic conditions in Wuhan, this paper points out the key monitoring section and establishes the monitoring steps in session 3. In order to support the conclusions in the paper, we have amended 3.1 Characteristics of cover-collapse sinkholes; deleted session 3.1.2 Distribution of limestone; and added a new session 3.1.4 Karst collapse susceptibility distribution.

The new structure of session 3 is:

3.1.1 Geological background

It introduces the geological background in Wuhan area. The added Figure 4 shows the distribution of lithology, tectonics and ground subsidence in the monitoring area.

3.1.2 Characteristics of karst collapse in Wuhan

It introduces the characteristics of karst collapse in Wuhan, which is the basis of the study of the warning criteria in PART 4 of this paper.

3.1.3 Previous karst collapse research in Wuhan

It introduces the research work of karst collapse in Wuhan. Monitoring and early warning research is a part of the whole work.

3.1.4 Karst collapse susceptibility distribution

It introduces the karst collapse susceptibility distribution in Wuhan, which is the basis of the selection of the monitoring sections.

*2. Figure 1. A geological sketch is needed in order to locate the study area, a continental view can help for the location against only China surface.*

Thank you for your comments, we have inserted Figure 1, to show the location of the research area in China; we have replaced Figure 4 Limestone distribution bands in Wuhan with geological map of karst monitoring area in Wuhan city on page 6.

*3. At figure 2 and 3, I assume that authors should have permission from both journals to include photographs from the cited manuscript, I suggest to editorial team to confirm this subject about the figure rights.*

Figures 2 and 3 have been replaced by using the Research Group's pictures in page 3.

*4. The cities referenced where sinkholes has been identified should be included in the map from Fig. 1.*

The sinkholes pit layer was added in the replaced Figure 4 on page 6.

*5. The text from page 4 (paragraph from line 15) "so by contrasting cross-sectional maps of the same traverse collected regularly over time by GPR, it is possible to estimate underground soil movement, and this helps to monitor the cave in terms of its formation and development, and thus enables prediction of cover-collapse sinkholes." Requires some references where this subject has been previously applied.*

GPR was employed to monitor Zhemu village in Guilin city, Guangxi by Li et al.(Li et al. (2005) (page 26, L29). This has been made clear on page 27, L14; and on page 4, L4.

*6. On the other hand, the evaluation of GPR in order to analyze karstic underground features should require to include, at least, some example of the obtained results and the compared evaluation of the same profiles during time to identify the changes related to the GPR record (besides the underground characteristics, soil state can also produce changes in the radargrams).*

Figure 2 shows the results of the Maotangang 1 # GPR monitoring profile in 2010 in Wuhan, Hubei Province. There are two anomalous reflection signals areas: (1) at the mileage of 170-180m and depth of 17.5m; (2) at the mileage of 240-255m and depth of 15m. Figure 3 shows the results of the same area in 2011. The comparison results show that the change at the second exception area is not obvious, but not the first one. It is indicated that the tendency of the depth developed to the surface is not exist. The found can be used as the evidence for the karst ground subsidence warning. Since the paper focused on the early warning system, so Figure 2 and Figure 3 did not included in the paper.

[Figure]

Figure 2 The results of Maotang 1 # GPR Monitoring Profile in 2010

[Figure]

Figure 3 The results of Maotang 1 # GPR Monitoring Profile in 2011.

*7. About the dynamic underground water level monitoring is difficult to understand as a generalization that collapse are produced by the pressure increase or the term "the cover layer will be damaged", this subject requires explanation.*

Lei et al. (1993) tested the karst ground subsidence model in the Yangzi River first terrace in Wuchang District where the paper studied area. The result shows that the mechanism of karst ground subsidence is mainly seepage deformation where is a dual structured covering layer of soil – the clayey is on the top of the sandy soil.    The saturated sandy soil located above the karst cavity is easily moved and formed the creep flow failure as long as the karst cavity water level decreases; and the damage will soon extend to the surface of the sandy soil, so that the upper clayey forms a cover layer. When the strength of the cover layer is not strong enough, the karst collapses occurred and the collapse process is shown in Figure 4.

The paper concluded that the mechanism of soil damage caused by the decline of karst water level is the soil seepage deformation (or subsurface erosion). In the model test, the soil damage critical hydraulic gradient is 12.89 for the single structured clayey covering layer of soil, and 2.03 for the dual structured covering layer of soil (the clayey is on the top of the sandy soil). When evaluating the risk of karst collapse in the study area, it can be done by comparing the maximum permeability hydraulic gradient (J) which is calculated from the pore water lever in the study area with the critical hydraulic gradient (Jc). When J > Jc, there are collapse risks.

[Figure]

Figure 4 The collapse formation process in the dual structured covering layer of soil

1 the pore water level      2 the karst water level

The amended part shows in the Line 13 to Linw22 in the page 10.

*8. About chapter 3.1.1. a geological map, section, borehole distribution or a cross-section should help in the interpretation of the context where the later analysis is carried out. If cross-section is included, the location of the water level should be also interesting to be included.*

The information was included in the newly replaced Figure 4 on page 6.

*9. Figure 4 also could require permissions from the journal from where it comes from, the aspect of the figure is not clear and besides the bands of carbonatic rocks, other geological subjects are required to be included. Moreover the bands are assumed to be limestones bands and not karstic bands as are referenced in the figure. Authors indicate that no evidences of karstic processes are in some of them. Moreover the reference to the structural setting, fold axis for example, requires being included in the figure.*

The new Figure 4 shows the results of the study on karst geologic conditions in monitoring area, mainly demonstrates the section of karst ground subsidence in Wuhan. In the newly added geological map of the study area, the lithology, geologic structure, and the distribution of sinkhole pits are included.

*10. The quality of fig. 5 is not evident, as there is not information that permits to contextualize it, moreover this does not make reference to risk, if there were anything evaluated in this figure should be hazard, peligrosity or susceptibility (not risk distribution).*

The research team used the AHP Method to evaluate the karst collapse susceptibility in Wuhan. Figure 5 is one of the consensuses on the risk of karst ground subsidence research in Wuhan. The purpose of putting in this paper is to illustrate the degree of the karst ground subsidence susceptibility in Wuhan. It is the susceptibility partition, not the risk partition. We are sorry the lack of grasp of the word, resulting in ambiguity.

*11. At figure group IV is incorrectly written and there is no sense about what "no suscetibility" means. The rest of the manuscript include description of what is presented to be done, with the considerations of the conclusions but there are not a data integration, comparison or discussion different than the general description. I think that what is presented can be of interest for the community but further work is needed including presentation, description and discussion of data. In this moment is more a general report of what to do, what can be expected and some results without the possibility to be evaluated by readers. In this sense, I suggest to modify significantly the manuscript before further revision.*

The term"no susceptibility" has been replaced with "non-Karst area" because the karst collapse is not exist in non-Karst area.

---

## Referee Comment (RC2) · Anonymous Referee #1 · 10 Apr 2017

i have had the opportunity to access to the answer from authors, but i have not been able to download the new manuscript version that points out to the modifications that have carried out. In this sense, as in the preceding manuscript version, the sound is good, but the evaluation of the methodology requires the possibility of contrasting the date. I consider that many of the pointed out subjects can be used in order to evaluate susceptiiblity to the karst phenomena, but i still have doubht about the possibility that they can be used as been pointed out as an early warning system.

---

## Referee Comment (RC3) · Anonymous Referee #1 · 13 Apr 2017

Dear editor and authors, this is also my first time with this review process type, I found it interesting while I expect to find a new version of the manuscript and not an answer to what can be do (I have not found the new version of the manuscript, then I cannot evaluate it or if the suggested changes and suggestions have been incorporated in the article). About the answer to my previous comments, manuscript is entitled with "early warning", that require a monitoring technique that permits to evaluate indicators that can be used for alert about a collapse that can appear.  In this sense, I understand the parts that make reference to evaluate the susceptibility, conceptually some of the proposed monitoring techniques can sound clear, but I am not sure that as presented and described can be used as an early warning.  Solution is a long process (fast in

geological terms), possibly the main part of the cavities can be present, or they change through long periods of time. The evaluation of water as a factor can produce the development of collapses due to the rheological changes related to water pressure change, weight, etc. Then this subject is of interest for monitoring but I am not sure if this is an early warning measurement as presented (e.g. about the evaluation of the rheological properties of the materials in a static manner). In some cases the topographical surficial change can be used as predictor for collapse, but it does not exclude that collapses can be used for this subject. GPR, for example, can be used to locate cavities, and to evaluate in time how it progress to surface, but the way to evaluate data does not permit to identify an early warning. All in all, my main concern is still the same related to the early warning, that is the main objective of the manuscript, being the rest of subject more related to susceptibility to karst processes that can be used in urban planning for example, I want to say, measurements to avoid the exposition to the hazard, not to live over it. The monitoring infrastructure that permits to avoid the hazard by early warnings is difficult to evaluate in the proposed manner because some of them requires to be continuously measured.

---

## Author Comment (AC2) · 13 Apr 2017

Responses to reviewer 1 comments 20160413
Ref: nhess-2017-222

We hope that the editor and reviewer 1 agreed with our assessment that the paper has been substantively improved

I have had the opportunity to access to the answer from authors, but i have not been able to download the new manuscript version that points out to the modifications that have carried out.

Sorry, because I'm not familiar with editing system, the modification process files are not sent, and I upload again now.

In this sense, as in the preceding manuscript version, the sound is good, but the evaluation of the methodology requires the possibility of contrasting the date. I consider that many of the pointed out subjects can be used in order to evaluate susceptibility to the karst phenomena, but i still have doubt about the possibility that they can be used as been pointed out as an early warning system.

Through the study of karst geological conditions in Wuhan , this paper makes a estimation on karst collapse susceptibility distribution in session 3. Finally, the evaluation result Figure 5 is obtained. Figure 5 is the basis which is selected for karst collapse key monitoring area.

The session 5 is the design of monitoring and early warning system. The session 5.4 designs the structure of the monitoring data stored in the on Page 16, L15. The session 5.5 is system warning modules which is calculated according to the monitoring data. It contains Hydraulic gradient warning module, Plastic zone warning module, GPR warning module and he Comprehensive warning calculation model.

The session 6.1 Monitoring data management is implementation of karst collapse monitoring data management capabilities in the system on the page 19, L9. All the monitoring data is managed in database established by using EXCEL software, which includes the data of soil pressure, land surface movement, cracks, karst water level, pore water level, monitoring points in wells, ground water level and rainfall level.

The session 6.2 Monitoring data warning level calculations is warning level calculations that the system calls relevant monitoring data in the EXCEL, containing Hydraulic gradient warning level calculation, Plastic zone warning level calculation, GPR warning level calculation and Determination of comprehensive warning levels.

---

## Author Comment (AC3) · 14 Apr 2017

[revised manuscript text omitted]

Wuhan area is located at the southern junction of the Yangtze Block and the Qinling-Dabieshan orogenic belt tectonically. The bedrock in Wuhan area is Silurian to Quaternary in age. The Silurian strata are shale and silty shale with thick-bedded siltstone interbedded. The Devonian strata are composed of medium-thick silicarenite and fine sandstone. The Carboniferous strata are mainly composed of silty claystone and bioclastic grainstone. The Permian are predominantly composed of carbonates and silicalites. The lithology of lower Triassic are predominantly composed of calcarenite and mudstone, while upper are composed of aubergine argillaceous siltstone and quartz sandstone. The lithology of Cretaceous-Paleogene are composed of a set of red clastic rock. The Quaternary deposits are widely distributed in the area, which include eluvium clay, diluvium clay, lacustrine silt clay and alluvial silt and sandy loam. Six folds form trough-like folds with wide anticlinal and narrow synclinal. These folds are successively named Fengheshan syncline, Daqiao overturned syncline, Xinlong-Baozixie compound overturned syncline, Zhuankou syncline and Dajunshan syncline from north to south in the area. Faults are developed which are mainly formed in Indosinian and Yanshanian period. NWW strike direction faults mainly contain Hengdian - Longkou fault (F1), Houhu - Baihushan fault (F2), Wudong fault (F3) and Longmiao fault (F4), which fault surface dip to the north with a dip angle of 35 - 60 degrees. Duoluokou fault (F5), Zhanjiaji - Jinkou fault (F6), Jiangjiadun - Qinglinghu fault (F7), Wutongkou-Tangxunhu fault (F8) and Yanxihu-Liufang fault (F9) are the representative faults, which strike direction are NNE.The bedrock in Wuhan is Silurian to Quaternary in age.

[revised manuscript text omitted]

**3.1.4 Karst collapse susceptibility distribution**

Based upon determinants of the karst collapse in Wuhan, we selected the karst foundation conditions (karst stratum, the developed degree of solution cavern, the distribution of karst sinkholes ), hydrogeological conditions (the relationship between pore water and karst groundwater, distance from the Yangtze River, whether it is affected by groundwater exploitation), geological structure (the influential degree of geological structure), upper cover conditions (thickness and structure of the covering layer) which include four factors with

[Figure]

a total of eight indicators as the Analytical Hierarchy Process combined with Information criteria in the karst collapse hazard model. This was used to assess the susceptibility of karst collapse in Wuhan, and the results are shown in Figure 5.

Fig.5. The karst collapse susceptibility distribution of monitoring area in Wuhan

**3.2 Monitoring karst collapse in Wuhan**

10 A project, "Wuhan cover-collapse sinkholes geo-hazard monitoring and early warning" was undertaken by the Hubei Geological Environment Station (HGES) as part of the project "Typical geological disaster monitoring and early warning and demonstration project", sponsored by the China Institute of Geo-Environment Monitoring (CIGEM) in 2006. By the end of 2008, HGES had established a cover-collapse sinkhole monitoring network in the high susceptibility karst collapse area, and has been monitored collapse since then. The selected monitoring

15 area is shown in red in Figure 4. Key monitoring areas are shown in Figure 5. According to the characteristics of karst collapse in Wuhan City, the karst collapse monitoring and early warning technology is mainly built on

three aspects: rock and soil movement, building deformation and groundwater change, together with analysis of geologically environmental conditions. The monitored content (Shen et al., 2014, Xu 2016) was as follows:

(1) Soil change monitoring

Two methods were used to monitor soil change: examining pressure meters buried in the ground in order to monitor pressure; tracking vibrating layers and cave cross-profiles from scan cross sectional profiles obtained by the GPR.

(2) Deformation monitoring

Deformation monitoring included land subsidence monitoring and land surface crack monitoring.

(3)  Groundwater monitoring

This included tracking groundwater levels in monitoring holes and wells. The former further includes examining the groundwater tables in Quaternary pore water and karst water.

**3.3 Cover-collapse sinkhole monitoring and warning implementation**

In order to monitor and to provide early warning of cover-collapse sinkholes, there were four critical elements: management of all types of monitoring data, development of warning criteria, measurement of severity of warning and warning classification.

(1) Warning classification

Three levels were used to classify the warning grade. Level I is "safe" indicated as green on the warning map; level II refers to the state "becoming dangerous", yellow; and level III "dangerous", red.

(2) Warning cell size

The cover-collapse sinkhole monitoring area was gridded at 50m × 50 m as warning units in Wuhan, with the warning signal shown in each unit in green, yellow and red colour accordingly. Grid units outside of the monitored area are displayed in light blue.

(3)  Monitoring and warning system implementation

(4)  The monitoring and early warning cover-collapse sinkhole system was further developed on a GIS software platform.

**4 Selection of warning criteria**

Based upon the cover-collapse sinkhole monitoring project and early monitoring knowledge, the current warning system selects data collected through GPR and underground water levels to determine the warning

classification.

**4.1 Hydraulic gradient warning criteria**

Lei et al. (1993) used a physical model of the first terrace of Yangtze River sediment in Wuchang District where is located in the paper study area and determined the mechanism of soil collapse derived from karst leakage

5    deformation caused by falling water level corroding limestone bedrock. The result shows that the mechanism of karst ground subsidence is mainly seepage deformation in where is a dual structured covering layer of soil – the clayey is on the top of the sandy soil. The saturated sandy soil located above the karst cavity is easily moved and formed the creep flow failure as long as the karst cavity water level decreases; and the damage will soon extend to the surface of the sandy soil, so that the upper clayey forms a cover layer. When the strength of the

10   cover layer is not strong enough, the karst collapses are occurred. The paper concluded that the mechanism of soil damage caused by the decline of karst water level is the soil seepage deformation (or subsurface erosion).  
[revised manuscript text omitted]

Xu, G. L.: Mechanism and Hazard Assessment of Covered Karst Sinkholes in Wuhan City, China, PH.D. , China University of Geosciences, Wuhan, 149 pp., 2016.

Yilmaz, I.: GIS based susceptibility mapping of karst depression in gypsum: A case study from Sivas basin (Turkey), Eng. Geol., 90, 89–103, doi:10.1016/j.enggeo.2006.12.004, 2007.

Zhao, D. J., Peng, F., Yang, J. and Song, Q.: Karst collapse hazard zoning evaluation based on AHP in Wuhan city, Chinese J. Res. Environ. Eng., 26, 97–99, 2012.

Zheng, X. Ch. and Wei, Zh. Y.: Analysis on induced factor of Karst Collapse in Wuhan City, Chinese J. Urban Geotech. Invest. Surv., 1, 15–19, 22, 2004.

Zhong Y., Zhang, M. K., Lan, L., Zho, Sh. K. and Hao, Y. H.: Risk assessment for urban karst collapse in Wuchang District of Wuhan based on GIS, Chinese J. Tianjin Normal University (Nat. Sci. Edition), 35, 49–53, 2015.

---

## Referee Comment (RC4) · Anonymous Referee #2 · 17 Apr 2017

The paper deals with cover-collapse sinkholes in China, with the goal to describe an early warning system for this type of phenomenon. The present version of the manuscript needs extensive work before being fully considered for publication. This derives from many reasons, that I will try to delineate in the following (with further comments, questions and suggestions in the attached file). First and foremost, the english language needs to be carefully checked by a English-native speaker. I tried my best to improve the english, but at several points (highlighted in the attached file) I simply could not catch the meaning of what the Authors were trying to convey. A check of the manuscript by a English-native speaker is mandatory to acceptance of the article. This should also be accompanied by a check of the terminology, which sometimes was not

properly used to me. In particular, at this latter regard (terminology), there is confusion about the use of the term risk. At many points, risk is used in an inappropriate way, with the actual meaning of susceptibility or of hazard. Authors must refer to the internationally accepted definition of risk, and use it in a correct way.

In general, the overall structure of the work is very poor, and causes great confusion in the reader. It is not easy to understand what of the manuscript is supported by real data, and real monitoring data are lacking. In my opinion, this cannot be considered as an early warning system, as stated by the Authors.

The geological background is very poor, and does not allow the reader to understand the situation in the study area. A geological map, with a schematic cross section, would be very useful at this aim.

Soil caves are mentioned in the paper, but there is no explanation of what they are. More details should be presented. I guess the Authors are talking about epikarst. In this case, they should better explain what they mean, also by quoting the wide available international literature on the subject.

A big drawback in the article is the lack of a clear geological-geotechnical model, which is crucial for understanding and explaining the type of sinkholes occurring, and for designing any possible system of prediction. A model is never presented in the article, which is a very weak point. At this regard, in reference to forecasting of sinkhole occurrence, see for instance Parise & Lollino (2011).

It is not clear to me how many of the references in the list are written in Chinese language. This should be clearly shown. I am afraid many of the papers are in Chinese. If this is true, the Chinese references should be kept at a minimum number, given they are not available to international readers. Further, the international literature deserves more attention, and at this aim I am suggesting a number of articles about sinkholes.

There are several problems with the figures. Many of them are missing the north, the

graphic scale, or both. In many others, at least the contour lines, or some elevation points of reference should be shown, in order to facilitate the reading. Others (for instance, figures 8 to 11) can be deleted, since they do not add anything to the content of the article.

In addition to those already indicated in the accompanying file, there are many references that should be cited by the Authors. These must be introduced, in order to better organize the article, and to put it in a more international context. Below I list some suggestions.

As regards karst hazard and management: Brinkmann R. & Parise M., 2012, Karst Environments: Problems, Management, Human Impacts, and Sustainability.An introduction to the Special Issue. Journal of Cave and Karst Studies, vol. 74 (2), p. 135-136. Gutiérrez F (2010) Hazards associated with karst. In: Alcántara I, Goudie A (Eds), Geomorphological Hazards and Disaster Prevention. Cambridge University Press, Cambridge, 161-175. North, L.A., van Beynen, P.E., Parise, M., 2009. Interregional comparison of karst disturbance: west-central Florida and southeast Italy. J. Environ. Manag. 9 (5), 1770–1781. Parise M., 2015, Karst geo-hazards: causal factors and management issues. Acta Carsologica, vol. 44 (3), p. 401-414. Parise M, Gunn J (Eds.) (2007) Natural and anthropogenic hazards in karst areas: Recognition, Analysis and Mitigation. Geol. Soc. London, sp. publ. 279. van Beynen, P.E., Townsend, K.M., 2005. A disturbance index for karst environments. Environ. Manag. 36 (1), 101–116.

As regards geophysical methods for cave information: Kaufmann G (2014) Geophysical mapping of solution and collapse sinkholes. J Applied Geophysics 111: 271–278. Kaufmann G, Romanov D, Nielbock R (2011) Cave detection using multiple geophysical methods: Unicorn cave, Harz Mountains, Germany. Geophysics 76 (3): 71–77. Margiotta S., Negri S., Parise M. & Valloni R., 2012, Mapping the susceptibility to sinkholes in coastal areas, based on stratigraphy, geomorphology and geophysics. Natural Hazards, vol. 62 (2), p. 657-676, DOI 10.1007/s11069-012-0100-1. Margiotta S., Negri

S., Parise M. & Quarta T.A.M., 2016, Karst geosites at risk of collapse: the sinkholes at Nociglia (Apulia, SE Italy). Environmental Earth Sciences, vol. 75 (1), p. 1-10, DOI: 10.1007/s12665-015-4848-y.

As regards epikarst: Jones W.K., 2013, Physical structure of the epikarst. Acta Carsologica, vol. 42 (2-3), p. 311-314. Williams P.W. (1983) The role of subcutaneous zone in karst hydrology. J. Hydrol., 61, 45–67. Williams P.W. (2008) The role of the epikarst in karst and cave hydrogeology: a review. Int. J. Speleol., 37 (1), 1–10.

As regards engineering problems in karst: Parise M., Closson D., Gutierrez F. & Stevanovic Z., 2015, Anticipating and managing engineering problems in the complex karst environment. Environmental Earth Sciences, vol. 74, p. 7823-7835, DOI :10.1007/s12665-015-4647-5. Waltham AC, Fookes PG (2003) Engineering classification of karst ground conditions. Quarterly Journal of Engineering Geology and Hydrogeology 36: 101-118. Zhou W, Beck BF (2011) Engineering issues on karst. In: P. van Beynen (Ed), Karst Management. Springer, Dordrecht, pp. 9-45.

As regards geological and geotechnical models for predicting sinkholes: Parise M. & Lollino P., 2011, A preliminary analysis of failure mechanisms in karst and man-made underground caves in Southern Italy. Geomorphology, vol. 134 (1-2), p. 132-143. Lollino P., Martimucci V. & Parise M., 2013, Geological survey and numerical modeling of the potential failure mechanisms of underground caves. Geosystem Engineering, vol. 16 (1), p. 100-112, DOI 10.1080/12269328.2013.780721.

For all the above reasons, I recommend major revisions for the manuscript in its present form.

Please also note the supplement to this comment: http://www.nat-hazards-earth-syst-sci-discuss.net/nhess-2017-22/nhess-2017-22-RC4-supplement.pdf
* * *
[Figure]

**Supplement:**

[revised manuscript text omitted]

---

## Author Comment (AC4) · 17 Apr 2017

Responses to reviewer 1 comments 20160414

Ref: nhess-2017-222

We hope that the editor and reviewer 1 agreed with our assessment that the paper has been substantively improved

Dear editor and authors, this is also my first time with this review process type, I found it interesting while I expect to find a new version of the manuscript and not an answer to what can be do (I have not found the new version of the manuscript, then I cannot evaluate it or if the suggested changes and suggestions have been incorporated in the article).

I am very sorry that I thought the new version of the manuscript had been uploaded at 12 April but in fact the upload is failed because the modification file is too large. Then I uploaded the new version of the manuscript again on April 14th.

About the answer to my previous comments, manuscript is entitled with "early warning", that require a monitoring technique that permits to evaluate indicators that can be used for alert about a collapse that can appear. In this sense, I understand the parts that make reference to evaluate the susceptibility, conceptually some of the proposed monitoring techniques can sound clear, but I am not sure that as presented and described can be used as an early warning. Solution is a long process (fast in C1 NHESSD Interactive comment Printer-friendly version Discussion paper geological terms), possibly the main part of the cavities can be present, or they change through long periods of time.

In the karst area, the bedrock collapse is that the bedrocks sink along the cavities and pipes under gravity. The soil collapse is that the overlying materials comprised mainly of the quaternary loose deposits over the carbonate rocks sink under gravity. According to the statistics for the karst collapse of ten southern provinces in China, the soil collapse accounted for 96.7 percent of the total. The karst collapse in this paper is the soil collapse, which is a phenomenon that the quaternary overlying materials collapse along the pipes to the cavities due to the effect of natural and human factors since there are cavities beneath the overlying strata and the pipes connecting the both of them. It's showed as Figure 1.

[Figure]

Figure 1 The process of karst collapse

Yellow: quaternary loose deposits    light blue: bedrock    green: groundwater table

Sometimes the speed of the karst collapse can be very fast. For example, the water inrush occurred in 2171 working faces of the Fangezhuang Mine of Kailuan (Group) Limited Liability Corporation at 10:20 am on June 2nd, 1984 and the entire mine was submerged after only about 21 hours. Moreover, 4 large-scale mines were damaged in different degree and the direct economic loss amounted to RMB 0.56 billion. In about 50 minutes after the water inrush occurred in the Fangezhuang Mine, the covered karst area from the Pengjiatatuo to the Linxi factory, about 3~7 km north of the water inrush point, collapsed due to sharp decline of the karst water table. There were 17 sinkholes with general diameters of 10 m and depths of 3~12 m, and then four of the sinkholes were connected together.    (Source: Wei Fenghua: Study in the Mechanism of Karst Collapse in Tangshan City, Chinese Journal of Geology and Prospecting, 2006,4(2), 86-89. )

The evaluation of water as a factor can produce the development of collapses due to the rheological changes related to water pressure change, weight, etc.
Then this subject is of interest for monitoring but I am not sure if this is an early warning measurement as presented (e.g. about the evaluation of the rheological properties of the materials in a static manner).

Three conditions needed to form a karst collapse are as follows:
(1) The cavities beneath the overlying strata and the pipes connecting both of them.
(2) The overlying materials comprised mainly of the loose deposits with a certain thickness.
(3) The hydrodynamic condition of the karst groundwater is easy to change.

The soil collapse is the most common karst collapse in Wuhan city. There is a close connection for surface water, phreatic groundwater and karst groundwater in the study area and the water exchange among them is completed by groundwater runoff. The hydrodynamic pressure induced by groundwater runoff is proportional to the hydraulic gradient. The hydraulic gradient increases due to the change of the water level in Yangtze River and the groundwater exploitation, which makes the groundwater flow faster and the hydrodynamic pressure increase. The hydrodynamic pressure will be greater than the cohesion and the friction of the soils, and the soils are carried away by the seepage which results in the formation of soil cavities in the overlying strata. The surface collapse occurs when the soil cavities expand to a certain stage.

Based on an investigation for the statistical relation between karst collapse and water-level fluctuation and change of hydraulic gradient, an early warning for karst collapse can be realized by using these two indexes.

There are some formation mechanisms of karst collapse, such as suffosion erosion theory, vacuum suction erosion theory, vibration theory, liquefaction theory and gas explosion theory et al. Furthermore, some scholars proposed the mechanism of soil rheology. At present, the suffosion erosion theory and vacuum suction erosion theory are recognized widely.

For the suffosion erosion theory, the overlying loose deposits and fillings in gaps are eroded and hollowed along pipes in bedrock by groundwater flow, which leads to the formation of soil

cavities and the collapse of roof.

For the vacuum suction erosion theory, in some shallow covered karst area with impermeable overlying strata, the pipes remain closed. Thus, a vacuum is formed between groundwater level and pipes when groundwater level falls fast below the bottom of soil. The overlying soil will be sucked by the vacuum which leads to a karst collapse. (source: Xu Weiguo, Zhao Guirong: The Implication of Suction Action for Ground Subsidence in Karst Mining Areas, Chinese Journal of Geological Review, 1981,27(2): 174-180.)

For the soil rheology theory, shear failure occurs easily inside saturated plastic-flowing soil due to pressure difference induced due to gravity and tension after groundwater level falls when the overlying stratum are mucky soil and soft clay. The cavities of soil expand continuously due to the soil erosion and the new relaxation zone formed continuously in a certain region of roof. As the cavities expand, the roof will finally collapse when the gravity of overlying strata is greater than the friction of surrounding soils. (source: Li Qianyin: Further study on formation mechanism of karst collapse, Chinese Journal of Geological Hazard and Control, 2009, 20,3, 52-55.)

In some cases the topographical surficial change can be used as predictor for collapse, but it does not exclude that collapses can be used for this subject.

Indeed, the monitoring programs for the karst collapse in Wuhan city involve the topographical surficial change. However, the topographical surficial change is not suitable to be a criterion for the karst collapse because the influencing factors of the topographical surficial change are too many.

GPR, for example, can be used to locate cavities, and to evaluate in time how it progress to surface, but the way to evaluate data does not permit to identify an early warning.

Indeed, the time that cavities progress to surface can be obtained by short period GPR. Because of the high cost of GPR, the monitoring period of GPR is 6 months in the monitoring programs for the karst collapse in Wuhan city, which is a kind of medium and long-term monitoring.

All in all, my main concern is still the same related to the early warning, that is the main objective of the manuscript, being the rest of subject more related to susceptibility to karst processes that can be used in urban planning for example, I want to say, measurements to avoid the exposition to the hazard, not to live over it.

With the economic development, the scale of Wuhan city is expanding fast. It is inevitable to construct in the covered karst area because the area of land is limited.

The monitoring infrastructure that permits to avoid the hazard by early warnings is difficult to evaluate in the proposed manner because some of them requires to be continuously measured.

Indeed, an early warning for karst collapse is on the basis of the continuous measure. So far, the

monitoring network for karst collapse in Wuhan city has been in operation for 8 years and the water-level fluctuation has been obtained successfully before the karst collapses occurred. Since 2009, the abrupt changes of water level have been monitored around the occurrence of 5 karst collapses. The water levels of monitoring holes (MT-1, CK13) in Maotangang increased abnormally in March, 2013. The water level of hole MT-1 increased abruptly on March 3, 5 and 7-12 and the maximal increase was 5.28 m as shown in Figure 2, which indicated that the karst collapses would occur in this area. A warning report was submitted in time to the Wuhan land resources and planning bureau from the Hubei Province geological environment Terminus and then 3 karst collapses occurred in this area from April 18th to 24th.

(source: http://www.chinatesting.com.cn/0hyzx/b/2015047493.html)

[Figure]

Figure 2 Time trend of M1-1 monitoring hole